# Spinal Reactive Oxygen Species and Oxidative Damage Mediate Chronic Pain in Lame Dairy Cows

**DOI:** 10.3390/ani9090693

**Published:** 2019-09-17

**Authors:** Daniel Herzberg, Pablo Strobel, Ricardo Chihuailaf, Alfredo Ramirez-Reveco, Heine Müller, Marianne Werner, Hedie Bustamante

**Affiliations:** 1Graduate School, Faculty of Veterinary Sciences, Universidad Austral de Chile, Valdivia 5110566, Chile; danielherzberg@gmail.com (D.H.); heine.jakob@gmail.com (H.M.); 2Animal Science Institute, Faculty of Veterinary Sciences, Universidad Austral de Chile, Valdivia 5110566, Chile; pablostrobel@uach.cl (P.S.); alfredoramirez@uach.cl (A.R.-R.); marianne.werner@uach.cl (M.W.); 3Department of Veterinary Sciences, Universidad Católica de Temuco, Temuco 4780000, Chile; rchihuailaf@uct.cl; 4Veterinary Clinical Sciences Institute, Faculty of Veterinary Sciences, Universidad Austral de Chile, Valdivia 5110566, Chile

**Keywords:** bovine, lameness, chronic pain, oxidative stress

## Abstract

**Simple Summary:**

Chronic inflammatory diseases could impact central nervous system homeostasis, being oxidative damage of the dorsal horn, a relevant mechanism mediating central sensitization. Chronic inflammatory lameness in dairy cows is a painful condition that affects animal welfare, affecting dairy production worldwide. This study reveals increased levels of reactive oxygen species, malondialdehyde, and carbonyl groups, indicating lipid and protein damage in the spinal cord of cows with chronic lameness. Moreover, antioxidant system activity was similar between lame and non-lame cows which suggests that antioxidant dysregulation was not the cause of oxidative damage, as has been proposed previously. Based on the fact that nociceptive pathways are strongly conserved between species, there is no reason to neglect that chronic pain in cows promotes Central Nervous System (CNS) alterations, such as oxidative damage. Moreover, lame cows develop central sensitization, as allodynia and hyperalgesia are centrally and not peripherally mediated. Our results support the current assumption that chronic pain is a central nervous system disease and lameness in dairy cows is far beyond an inflammation of the hoof.

**Abstract:**

Lameness in dairy cows is a worldwide prevalent disease with a negative impact on animal welfare and herd economy. Oxidative damage and antioxidant system dysfunction are common features of many CNS diseases, including chronic pain. The aim of this study was to evaluate the levels of reactive oxygen species (ROS) and oxidative damage markers in the spinal cord of dairy cows with chronic inflammatory lameness. Locomotion score was performed in order to select cows with chronic lameness. Dorsal horn spinal cord samples were obtained post mortem from lumbar segments (L2–L5), and ROS, malondialdehyde (MDA), and carbonyl groups were measured along with the activity of superoxide dismutase (SOD), catalase (CAT), glutathione peroxidase (GPx), and total antioxidant response (TAR). Lame cows had increased levels of ROS, MDA, and carbonyl groups, while no differences were observed between lame and non-lame cows in SOD, GPx, CAT, and TAR activity. We conclude that painful chronic inflammatory lameness in dairy cows is associated with an increase in ROS, MDA, and carbonyl groups. Nonetheless, an association between ROS generation and dysfunction of the antioxidant system, as previously proposed, could not be established.

## 1. Introduction

Chronic pain is considered a Central Nervous System (CNS) disease [1,2]. Central sensitization, an inherent feature of chronic pain, has been associated with a reduced neuronal threshold and increased activity that enhances nociceptive input to supraspinal levels [1]. It has been established that central sensitization in the spinal cord in experimental models of neuropathic and inflammatory pain is mediated by reactive oxygen species (ROS) and reactive nitrogen species (RNS) [3,4]. Increased amounts of free radicals (FR) associated with oxidative stress have been found during central sensitization in the spinal cord in neuropathic and inflammatory pain models. FR promote central sensitization through various mechanisms, including increasing phosphorylation of the NR1 subunit of the N-methyl-D-aspartate (NMDA) receptor [5], inhibition of the gamma-aminobutyric acid (GABA) transmission [6], and activation of the transient receptor potential cation channel subfamily V member 1 (TRPV1) channels [7]. Furthermore, FR can induce glial activation [8], excitotoxicity, cytokine release, and neuroinflammation [9,10]. Additionally, the role of oxidative stress during chronic pain states has been confirmed through the administration of FR scavengers, which significantly attenuates chronic pain behavior in humans and experimental models [11,12].

FR scavengers maintain an adequate balance between free radicals’ production and elimination [13]. For example, SOD accelerates the reaction of superoxide (O_2_^.-^) to form hydrogen peroxide (H_2_O_2_) and oxygen (O_2_), while glutathione peroxidase (GPx) reduces H_2_O_2_ into water, and lipids hydroperoxide into alcohols. Other scavengers of importance are catalase, which decomposes H_2_O_2_ into water; thioredoxins, which reduce oxidized proteins, and peroxiredoxins, which regulate H_2_O_2_ levels [14]. Nonetheless, in neuronal tissue FR can induce protein modifications, thus inactivating antioxidant capacity and enhancing oxidative stress.

Naturally occurring painful diseases have recently gained attention based on their potential to complement the results obtained from experimental pain models [15]. Accordingly, transitional pain models could increase the development of new analgesic compounds [16]. Currently, different chronic painful diseases in animals considered to be transitional include osteoarthritis in dogs and cats [17,18]. Interestingly, the potential use of chronic pain conditions in large animals as a translational model has not been studied in detail [19].

Lameness in dairy cows is a highly prevalent disease that severely affects animal welfare [20]. Additionally, lameness-associated pain causes a negative economic impact, reducing milk production, reproductive index, among others [21,22]. The inflammatory response associated with the hoof lesion is the primary event that leads to chronic pain [23]. Many features of chronic painful lameness in dairy cows resemble chronic pain in other species, including humans, including chronicity and comorbidity development [24,25]. Claw lesions promote histological changes in bone tissue, which is often observed in chronic painful diseases in humans. Recently, an interesting radiographic analysis of claw confirmed these findings [26]. Additionally, thermographic analysis of hooves could facilitate early identification of lameness [27,28]

Previous studies have reported a systemic increase in oxidative stress biomarkers in dairy cows which has been associated with hoof lesions and lameness [29,30]. Nonetheless, the pro-oxidative status of the spinal cord in cows with chronic inflammatory lameness has not yet been elucidated. The aim of this study was to investigate the redox status in the spinal cord and oxidative damage markers in cows with chronic pain associated with inflammatory lameness.

## 2. Materials and Methods

### 2.1. Bioethics Statement

The experimental protocol was approved by the Ethics Committee of Animal Research of the Universidad Austral de Chile (NO 323/2018).

### 2.2. Animals

Four lame cows were selected from a dairy farm, and six control animals were selected from a local slaughterhouse. Animals were Friesian and kiwi crossed with a range of parity between 2 and 6. The inclusion criteria for the animal selection of lame animals included a history of hind limb lameness of at least five months, with lameness being one of the most prevalent diseases in Southern Chile (e.g., white line disease, sole hemorrhage, sole ulcer, and digital dermatitis). Cows were euthanized after general intravenous anesthesia by administering an intrathecal injection of lidocaine in the atlanto-occipital foramen (farm animals) or according to the national regulations by mechanical stunning and exsanguination (slaughterhouse animals).

### 2.3. Lameness Assessment

Cows were classified into two groups: Lame (n = 4) or Non-lame (n = 6). Lameness was confirmed and classified according to the mobility score previously described [31]. The exclusion criteria for both groups included the presence of visible acute wounds, visible neurological gait alterations (central or peripheral ataxia), and acute or chronic mastitis.

### 2.4. Spinal Cord Processing, Protein Extraction, and Quantification

Immediately post mortem, a 20 cm segment of the spinal cord (L2–L4) was aseptically obtained after removal of the dorsal aspect of the lumbar vertebrae dorsal laminae. Dura mater and arachnoids meninges were gently dissected and washed with phosphate-buffered saline (PBS). Samples were sectioned and snap-frozen in liquid nitrogen and transported to the laboratory for further processing. Spinal cord segments of approximately 250 mg were homogenated in 1 mL of PBS using an Ultra Turrax tissue homogenizer (T10, IKA^®^, Staufen, Germany) at 16,000 rpm three times for 30 s each at 4 °C. Samples were then centrifuged at 2000 g for 3 min, and the supernatant was collected. Protein quantification was performed using the Pierce^TM^ bicinchoninic acid (BCA) protein assay kit (Thermo Scientific, Rochford, USA).

### 2.5. Laboratory Analysis

The following indicators of redox status were analyzed ROS, MDA, carbonyl groups, total antioxidant potential and SOD, CAT, and GPx activity. All analysis was performed in triplicates.

ROS concentration was determined according to the method previously reported by Gao et al. [32]. Briefly, 100 μL of supernatant were mixed with 10 μL of 0.5 mM 2’, 7’-dichlorofluorescin diacetate (DCFH_2_-DA) (Sigma-Aldrich, Santiago, Chile) and loaded into a 96-well plate and incubated at 37 °C for 30 min. Samples were read using a fluorometry microplate reader with an excitation wavelength of 500 nm and an emission wavelength of 525 nm. ROS levels were expressed as fluorescence/mg of protein.

MDA levels were measured using the thiobarbituric acid (TBARS) assay, as previously described by Ohkawa et al. [33]. Briefly, 70 μL of supernatant were mixed with 150 μL of 0.8% thiobarbituric acid (TBA) (Sigma-Aldrich, Santiago, Chile) and 150 μL of 20% acetic acid Sigma-Aldrich, Santiago, Chile). Samples were then incubated at 95 °C for 45 min. After cooling at room temperature (18–20 °C), samples were mixed with a 500 μL of butanol-pyridine (15:1, p/v) (Sigma-Aldrich, Santiago, Chile) and rocked for 30 s. The butanol layer was then separated by centrifugation at 10,000 rpm for 3 min, and quantitation of the organic layer was performed measuring absorbance at 540 nm. MDA concentration was expressed as μmol/gr of protein.

Carbonyl groups were measured according to that previously reported by Mesquita et al. [34]. Briefly, 2,4-dinitrophenylhydrazine (DNPH) (Sigma-Aldrich, Santiago, Chile) was used to conjugate carbonyl groups in an alkaline medium. The molar extinction coefficient of carbonyl conjugated complex was used to calculate its concentration in a microplate reader Sunrise (Tecan), and the concentration was expressed as nmoles/mg of protein.

Total antioxidant response (TAR) was measured according to that previously reported [35]. Briefly, hydroxyl radicals were produced using the Fenton reaction by mixing a standard solution of ortho-dianisidine/Fe^++^ with a standard solution of H_2_O_2_. Ortho-dianisidine undergoes oxidation into dianisidyl radicals yielding to a bright yellow-brown color compound. Antioxidants in the sample suppress this color formation proportional to their concentration. The rate of this reaction was measured at 444 nm, and the result was expressed in mmol equiv Trolox/L.

Superoxide dismutase (SOD; EC. 1.15.1.1) activity was measured using the reactive kit RANSOD (Randox, Crumlin, UK) at 37 °C. The method used the xanthine-xanthine oxidase complex to produce superoxide radicals that reacted with feniltetrazolium. The absorbance of this reaction was measured at 560 nm, and the enzymatic activity was expressed as U/mg of protein.

Catalase (CAT) activity was measured at 37 °C, according to Hadwan and Ali [36]. For this, ammonium vanadate reacts with hydrogen peroxide in an acidic solution, forming a peroxovanadium complex which is inversely proportional to CAT activity. The reaction was measured at 452 nm, and activity was expressed as U/mg of protein.

Glutathione peroxidase (GPx; EC. 1.11.1.9) activity was measured using the RANSOD (Randox, Crumlin, UK) kit at 37 °C, according to that previously reported [37]. Briefly, GPx catalyzes the oxidation of glutathione in the presence of tert-butyl hydroperoxide. Glutathione reductase reduces oxidized glutathione in the presence of NADPH, while NADPH is oxidized to NADP. The reaction was measured at 340 nm, and its activity was expressed as U/mg of protein.

### 2.6. Statistical Analysis

The normality of the data and variance homoscedasticity were evaluated using the Kolmogorov–Smirnov and the Shapiro–Wilk test, respectively. Accordingly, differences between lame and non-lame animals for each variable were evaluated using the *t*-test. A *p*-value of less than 0.05 was considered significant.

## 3. Results

Lame cows had higher ROS (350.5 ± 75.22 fluorescence/mg of protein) compared to non-lame cows (152.5 ± 28.62 fluorescence/mg of protein) (*p* < 0.05) (Figure 1A). Similarly, the thiobarbituric acid reaction product was increased (*p* < 0.01) in lame cows compared to non-lame (1.23 ± 0.2 versus 0.52 ± 0.06 µmol/gr of protein), indicating an increase in MDA (Figure 1B). The carbonyl groups concentration in the spinal cord of lame cows were higher (*p* > 0.05) than control cows (8.9 ± 3.9 versus 3.5 ± 1.6) (Figure 1C). A numeric, non-significant increase in SOD, and CAT activity was observed in lame cows (Figure 2A,B). In contrast, lame cows showed a numeric and non-significant decrease in GPx activity and TAR compared to control cows (Figure 2C,D).

## 4. Discussion

Increased levels of ROS were observed in the spinal cord of lame cows (Figure 1). Nociceptive stimulation increases metabolic rate and ROS production in the spinal cord, which might explain one possible source of spinal ROS in lame cows [38]. Concomitant to ROS increase, lame cows demonstrated higher levels of lipid and protein oxidation markers. Similar to our findings, several previous studies using inflammatory and neuropathic pain models have described a potential association between ROS and chronic pain [3,11,12]. Also, central sensitization leading to chronic pain maintenance has been associated with molecular changes and proteome modifications in the spinal cord and peripheral nerve [39,40,41,42]. These changes in the spinal cord lead to protein synthesis and protein folding leading to ROS generation [43]. Oxidative signaling regulates various molecular mechanisms involved in central sensitization, especially those mediated by phosphorylation, such as activation of Protein Kinase C (PKC) and NMDA receptor [5]; TRPV1 channels [7], along with inhibition of GABAergic transmission [6]. Moreover, mitochondrial and endoplasmic reticulum (ER) ROS generation has been linked to ER stress and an unfolded protein response (UPR) in the dorsal root ganglion and spinal cord [44,45]. These responses have been recently shown to be involved in both inflammatory and neuropathic chronic pain [46,47,48]. In our study, ROS determination was performed using the 2’, 7’-dichlorofluorescin diacetate (DCFH_2_-DA) probe, which has been described as a suitable method for measuring intracellular ROS production [49,50]. DCFH_2_-DA has been previously used to determine ROS levels in the spinal cord of rats after experimental trauma [51] and in the plasma of cows with chronic lameness [52]. A limitation of using DCFH_2_-DA, is its lack of specificity, as it gets reduced by O_2_^.-^, H_2_O_2_, -OH and by peroxynitrites [49,50]. Based on this, the oxidation of DCFH_2_-DA must be used only as an indicator of oxidative stress and not as a specific ROS marker [49,50].

Malondialdehyde is a lipoperoxidation product that has been extensively used as a biomarker of lipid peroxidation and oxidative damage [53]. Experimental pain models have shown increased levels of MDA in the spinal cord and sciatic nerve [54]. Similarly, chronic pain attenuation after antioxidant treatment has been associated with a reduction in MDA and other lipoperoxidation products in the spinal cord [55,56]. Plasma levels of MDA have also been studied as an indicator of oxidative stress and pain. MDA increases in the plasma of cows with chronic inflammatory lameness [29] and human patients with low back pain and rheumatoid arthritis [54]. Moreover, in humans, an improvement in motor function and pain relief was associated with lower levels of plasma MDA [57]. Nonetheless, the mechanism by which MDA promotes maintenance of pain has not been elucidated yet. However, lipid peroxidation products can easily diffuse across membranes and alter protein structure, acting as second messengers [53]. Moreover, one of the main mechanisms of MDA-mediated damage is its capacity to form highly immunogenic adducts [58], such as malondialdehyde-acetaldehyde, which has been proven to induce autoantigen synthesis promoting pain and inflammation in humans’ patients with osteoarthritis [59].

Carbonyl groups were also significantly increase in the spinal cord of lame cows. Carbonyl groups are frequently used as indicators of protein oxidation and are considered consistent markers of irreversible oxidative damage [60]. Moreover, lipids, DNA, and proteins could strongly bind to carbonyl groups [61]. Carbonylated proteins cannot be repaired by cellular enzymes, and their accumulation impairs protein function [62], promotes protein aggregates [63], and activates several signaling pathways [64]. Carbonyl modification could activate an inflammatory response. A nexus has been established between carbonylation of thiol groups in thioredoxins and NF-κB activation, nuclear migration, and expression of pro-inflammatory genes [65]. Certain carbonylated proteins could also be recognized as damage-associated molecular patterns (DAMP’s) by pattern recognition receptors (PRR) in order to promote the immune response [66]. Similarly, the potential role of carbonyl groups in the development of several human diseases including cardiac failure, sepsis, chronic renal failure, chronic lung disease, and Alzheimer’s disease has been described [62]. In the spinal cord, protein carbonylation increases in rats with autoimmune encephalitis (EAE). Moreover, protein carbonylation has been shown to increase in the cerebrospinal fluid of human patients with demyelinating diseases [67]. It’s been suggested that carbonyl groups could also bind to astrocytes promoting glutamate excitotoxicity by interfering with glutamate reuptake from the synaptic cleft [68], and thus, affecting astrocyte function [69]. Similarly, an increased number of carbonyl groups have been detected in astrocytes after traumatic brain injury (TBI) and EAE [68]. This mechanism mediated by oxidative stress could be of increasing importance, given that glutamate reuptake inhibition is an important pathway to central sensitization maintenance and chronic pain [70].

An interesting finding here reported is that despite the increased levels of lipid and protein damage, no evidence of antioxidant system dysfunction in the spinal cord of lame cows was found. Superoxide dismutase, CAT, GPx, and TAR activity in lame cows were similar to that observed in non-lame cows. Contrary to our findings, a reduction in the activity of the spinal mitochondrial SOD (Mn-SOD) after intraplantar injection of formalin in rats has been described [11]. SOD activity was reduced in the spinal trigeminal nucleus after an experimental model of facial inflammatory pain [71]. Some authors report that antioxidant system dysfunction mediated by Mn-SOD inhibition is one of the main mechanisms by which free radicals increase and promote central sensitization [4,11]. This finding has been previously confirmed using compounds that mimic SOD activity, including phenyl N-tert-butylnitrone (PBN), 4-hydroxy-2,2,6,6-tetramethylpiperidine 1-oxyl (Tempol), M40403 and M40401, which drastically and transitory reduced pain [11,12,72]. However, it is possible that spinal antioxidant dysfunction might not be the main mechanism of ROS generation in the lame cows at the time we performed the study. Our results suggest that ROS can promote oxidative damage and pain regardless of the state of the antioxidant system. Accordingly, [54] reported a marked increase in spinal SOD activity concomitant to increased oxidative damage in the spinal cord of rats with neuropathic pain. Similarly, Guedes et al. [73] described increased GPx activity in the spinal cord of rats after sciatic nerve transection.

Total antioxidant potential (TAR) was evaluated in order to analyze the complete antioxidant activity present in the spinal cord of lame cows, and to the authors’ knowledge, there is no previous report of spinal TAR measurement in experimental pain models. However, TAR has been evaluated in serum and plasma of human patients with persistent pain [74,75]. TAR in patients with chronic migraine was not different from controls, despite a significant increase in oxidative DNA damage detected in patients with pain [75]. In contrast, serum TAR has confirmed a negative correlation between pain intensity and antioxidant activity in human patients with fibromyalgia [74]. Moreover, plasma TAR has been evaluated in dairy cows in order to identify a possible association between oxidative status and lameness development during the peripartum. Nonetheless, similar to our findings, no significant difference between non-lame and lame cows was found [52].

Some limitations in our study include a low number of lame cows, which can increase the chance for beta error associated, decreasing the power of a statistical test. Nonetheless, posthoc power analysis was performed revealing that the statistical power fluctuated between 89 and 92%. Additionally, newer and more sophisticated techniques, such as gas chromatographymass spectrometry (GC-MS/MS), and liquid chromatography-mass spectrometry (LC-MS/MS) could increase the accuracy of the measurement of oxidative variables.

## 5. Conclusions

The results presented confirm that painful chronic inflammatory lameness in dairy cows is partially associated with an increase in ROS, lipoperoxidation products, and irreversible posttranslational proteins modification mediated by carbonyl groups. Nonetheless, an association between ROS generation in the spinal cord of lame cows and dysfunction of the antioxidant system as previously proposed could not be established.

## Figures and Tables

**Figure 1 animals-09-00693-f001:**
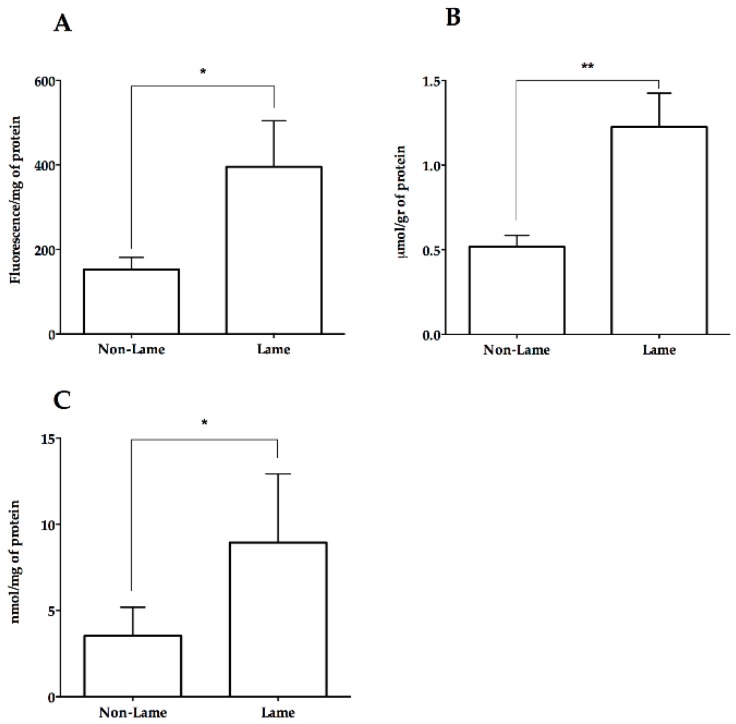
Spinal concentration of reactive oxygen species (**A**), malondialdehyde (**B**), and carbonyl groups (**C**) in experimental cows with chronic inflammatory lameness. * *p* < 0.05, ** *p* < 0.01.

**Figure 2 animals-09-00693-f002:**
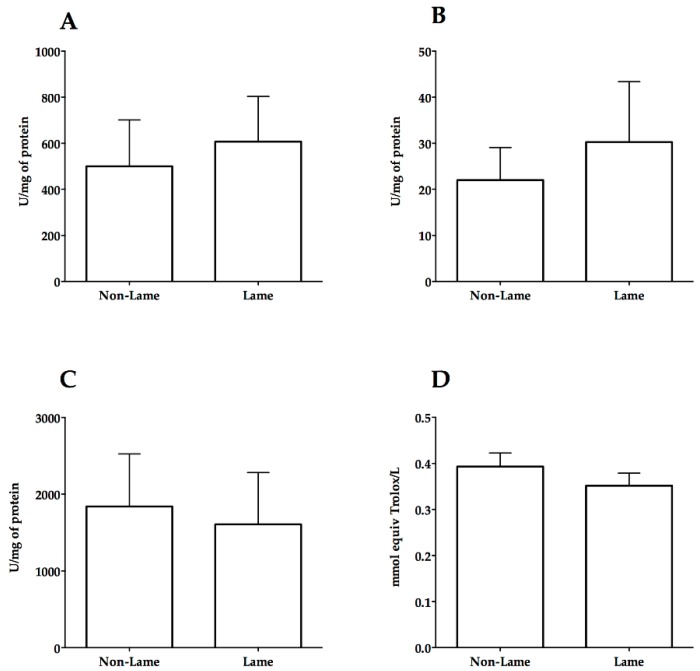
Spinal activity of superoxide dismutase (**A**), catalase (**B**), glutathione peroxidase (**C**), and total antioxidant response (**D**) in experimental cows with chronic inflammatory lameness.

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
