# Peer review of "Spinal Reactive Oxygen Species and Oxidative Damage Mediate Chronic Pain in Lame Dairy Cows"

_animals, 2019, doi:10.3390/ani9090693_

Round 1
Reviewer 1 Report
The manuscript describes interesting work. Though the manuscript is well written and has scientific benefits, still some improvement and clarifications, main in the introduction and methodology part are required. See the following comments in attached file for more specific suggestions.

Author Response
REVISION NOTE FOR REVIEWER 1
Lines 68-72: I suggest you add and develop in Introduction:
Fiore E., Perillo L., Marchesini G., Piccione G., Giudice E., Zumbo A., Armato L., Fabbri G., Gianesella. M. (2019). Effect of Parity on Claw Horn Lesions in Holstein Dairy Cows: Clinical and Radiological Study. Ann. Anim. Sci., 19( 1), 147–158.
Novotna I., Langova L., Havlicek Z. (2019). Risk Factors and Detection of Lameness Using Infrared Thermography in Dairy Cows – A Review. Ann. Anim. Sci., 19(3), 563–578.
Action: We have included these references in the introduction as requested by the reviewer.
Line 85: That is a very imprecise statement. Please state precisely the age of the cows and the breed.
Action: We have included parity range and breeds of the animals from which we obtained the spinal cord samples.
Line 93: Please state how many cows were in each group: Lame (4 cows?) or Non-lame (6 cows?)
Action: Modified
Lines 94-96: I think it's unnecessary - there were only Lame (severly lame) or Non Lame in the group. It suggests deleting the sentence.
Action: Modified
Line 149: Shapiro-Wilk
Action: Modified
Lines 191-195: This information should be included in M&M. Please remove it from here.
Action: We have removed the phrase as suggested by the reviewer
Lines 278-279: Please specify these techniques or add quotations for studies using such techniques.
Action: We have included these techniques as requested by the reviewer.
Line 292: Add the new references.
Action: We have included the new references in the references list
Figures 1 and 2: ... in experimental cows. Drawing signature needs to be corrected.
Action: Modified
Reviewer 2 Report
Since only 4 animals are in the case group and 6 animals in the control group, the beta error has to be addressed very carefully otherwise the manuscript is not scientifically sound. Also the conclusion consists of potential error which has to be adressed. I encourage the authors to correct this points an let the paper re-review.
Specific points are in the attached list

Author Response
REVISION NOTE FOR REVIEWER 2
Lines 23: The conclusion is not adequate, if CNS and Leg are independent from each other:
Action: Several lines of research support the neurodegenerative effects that chronic pain exerts over CNS. ROS exacerbation and oxidative damage is currently considered a hallmark of neurodegenerative diseases and several peripheral inflammatory pain models demonstrate increased ROS production and oxidative damage in the spinal cord. Based on the fact that nociceptive pathways are strongly conserved between species, there is no reason to neglect that chronic pain in cows promote CNS alterations, such as oxidative damage. Even more, the fact that lame cows included in our study showed marked hyperalgesia behavior is also a clear demonstration of central sensitization, as allodinya and hyperalgesia are centrally and not peripherally mediated.
Line 35: See simple summary
Action: We believe this in answered in the previous response
Line 151: As you only used 4 cases and 6 controls, the power has to be measured. A power of <0,8 (80%) shows that the results even significant are not trustful to potential beta error
Action: We performed power analysis previous to the start of the study. Additionally, we performed a post hoc power analysis using the G.Power Software including 6 and 4 as N por each group with an alpha value of 0.05 and including a 30% standard deviation from each mean that we calculated from our variables. We performed this analysis for every variable and we calculated a power that fluctuated between 89 and 92%. It is important to mention that the 30% value for the SD was higher than those obtained in our results.
Lines 204: folding?
Action: Folding refers to protein folding which is the molecular process by which a protein folds into tertiary or quaternary structure
Line 251: The conclusion is not adequate, if CNS and Leg are independent from each other, this results has to be explained differently. With your low number of animals this is very important due to a possible beta error or 2nd class error. You have to discuss this very carefully and your new conclusion must be also presented in the summary above.
Action: Somatosensory and especially in the nociceptive pathway we cannot express that CNS and Leg are independent from each other. Please refer to the first response in this revision note.
Lines 264: Discuss here the possible beta error
Action: We have included the potential beta error in the limitation paragraph which also mentions the low number of animals in each group.
Lines 279: Explain what differences could be addressed as significant with only 4 cases and 6 controls
Action: We believe that we have discussed the low number of animals in the limitation section before the conclusion.
Line 285: Maybe results have to be relativated if the power is less than 0,8
Action: We believe that we have addressed this concerns in the previous responses.
Round 2
Reviewer 2 Report
Lines 23: The conclusion is not adequate, if CNS and Leg are independent from each other:
Action: Several lines of research support the neurodegenerative effects that chronic pain exerts over CNS. ROS exacerbation and oxidative damage is currently considered a hallmark of neurodegenerative diseases and several peripheral inflammatory pain models demonstrate increased ROS production and oxidative damage in the spinal cord. Based on the fact that nociceptive pathways are strongly conserved between species, there is no reason to neglect that chronic pain in cows promote CNS alterations, such as oxidative damage. Even more, the fact that lame cows included in our study showed marked hyperalgesia behavior is also a clear demonstration of central sensitization, as allodinya and hyperalgesia are centrally and not peripherally mediated.
Reviewer: The action to lines 23 have to be implemented into the text for better understanding to the reader.
------
Line 151: As you only used 4 cases and 6 controls, the power has to be measured. A power of <0,8 (80%) shows that the results even significant are not trustful to potential beta error
Action: We performed power analysis previous to the start of the study. Additionally, we performed a post hoc power analysis using the G.Power Software including 6 and 4 as N por each group with an alpha value of 0.05 and including a 30% standard deviation from each mean that we calculated from our variables. We performed this analysis for every variable and we calculated a power that fluctuated between 89 and 92%. It is important to mention that the 30% value for the SD was higher than those obtained in our results.
Reviewer: The action to line 151 has to be implemented into the text for better understanding that the results presented in the text have scientific quality.
---------
Lines 204: folding?
Action: Folding refers to protein folding which is the molecular process by which a protein folds into tertiary or quaternary structure
Reviewer: This was clear to the reviewer but please write just: protein folding.
Author Response
REVISION NOTE FOR REVIEWER 2
Reviewer: The action to lines 23 have to be implemented into the text for better understanding to the reader.
Action: We have included the reviewer suggestion in the simple summary (Lines 22 to 25).
Reviewer: The action to line 151 has to be implemented into the text for better understanding that the results presented in the text have scientific quality.
Action: We have included the results of our power analysis in the limitation paragraph as suggested by the reviewer (Lines 282 to 283).
Reviewer: This was clear to the reviewer but please write just: protein folding.
Action: Modified according to the reviewer suggestion (Line 206).